# A Proposal for Botulinum Toxin Type A Injection Into the Temporal Region in Chronic Migraine Headache

**DOI:** 10.3390/toxins12040214

**Published:** 2020-03-28

**Authors:** Young-gun Kim, Jung-Hee Bae, Hyeyun Kim, Shuu-Jiun Wang, Seong Taek Kim

**Affiliations:** 1Department of Orofacial Pain & Oral Medicine, Yonsei University College of dentistry, Seoul 03722, Korea; butterbow@gmail.com; 2Department of Oral Medicine, Sun Dental Hospital, Sun Medical Center, Daejeon 34813, Korea; 3Department of Dental Hygiene, Division of Health Sciences, Namseoul University, Cheonan 31020, Korea; jung18342@naver.com; 4Department of Neurology, Catholic Kwandong University College of Medicine, International St. Mary’s Hospital, Incheon 22711, Korea; imkhy77@gmail.com; 5Department of Neurology, Neurological Institute, Taipei Veterans General Hospital, Taipei 112, Taiwan; sjwang@vghtpe.gov.tw; 6Faculty of Medicine, National Yang-Ming University School of Medicine, Taipei 112, Taiwan

**Keywords:** chronic migraine, auriculotemporal nerve, Botulinum toxin injection

## Abstract

Botulinum toxin type-A (BTX-A) injection for treating chronic migraine (CM) has developed into a new technique covering distinct injection points in the head and neck regions. The postulated analgesic mechanism implies that the injection should be administered to sensory nerves rather than to muscles. This study aimed to determine the topographical site of the auriculotemporal nerve (ATN) and to propose the effective injection points for treating CM. ATNs were investigated on 36 sides of 25 Korean cadavers. The anatomical structures of the ATN were investigated focusing on the temporal region. A right-angle ruler was positioned based on two clearly identifiable orthogonal reference lines based on the canthus and tragus as landmarks, and photographs were taken. The ATN appeared superficially in the anterosuperior region of the tragus. The nerve is located deeper than the superficial temporal artery. And it runs between the artery and the superficial temporal vein. In the superficial layer, it is divided into anterior and posterior divisions. The anterior division runs in a superior direction, while the posterior division runs in front of the ear and the several branches are distributed to the skin. We suggest that the optimal BTX-A injection points for CM are in the temporal region. The first point is about 2 cm anterior and 3 cm superior to two orthogonal reference lines defined based on the tragus and canthus, and the second point is about 4 cm superior to the first point. The third and fourth points are recommended about 2 cm superior to the first point, but respectively 1 cm anterior and posterior to it.

## 1. Introduction

Chronic migraine (CM) is a disabling neurologic disorder that affects 1.4–2.2% of the general population [1]. The beta version of the Third Edition of the International Classification of Headache Disorders (ICHD-3) defines CM as headaches occurring on at least 15 days per month for longer than 3 months, with the features of migraine headache occurring on at least 8 days per month [2]. CM is the most common type of primary chronic daily headache, and it is difficult to treat. OnabotulinumtoxinA (BOTOX^®^, Allergan, Irvine, CA, USA) was approved in 2010 by regulatory agencies in the USA for the treatment of CM, and it is still the only prophylactic therapy specifically approved for CM.

The analgesic effects of botulinum toxin type-A (BTX-A) were first reported in 1985, based on a pilot study of BTX-A treatment for cervical dystonia, which is characterized by abnormal and involuntary neck and shoulder muscle contractions that often result in significant and disabling musculoskeletal pain [3]. Several subsequent researches have investigated the effect of BTX-A in relieving numerous pain conditions, including myofascial pain syndrome, blepharospasm, temporomandibular disorder and bruxism, back pain, painful myoclonus, prostatic pain, cluster headache, tension-type headache, and migraine headache [4,5,6,7,8,9,10,11]. The association between BTX-A and pain relief was originally thought to be related only to its effect on muscle contraction. However, the results from several studies suggest that muscle relaxation effects do not directly coincide with the relief of pain, implying that other mechanisms underlie the analgesic effects of BTX-A [12]; for example, the analgesic effect of the toxin often occurs earlier and lasts longer than its effect on muscle tone. There is evidence that BTX-A affects afferent transmission [13,14] and inhibits the release of substance P, which plays roles in pain perception, vasodilation, and neurogenic inflammation [15]. Moreover, BTX-A may block the release of the calcitonin-gene-related neuropeptide (CGRP) and glutamate [16]. It can therefore be proposed that the analgesic effects of BTX-A is related to inhibition of pain transmission in sensory neurons, rather than its effects in motor neurons. 

BTX-A injections for treating chronic migraine (CM) has developed into a new technique covering distinct injection points in the head and neck regions [17]. That study suggested that BTX-A should be injected intramuscularly across seven head and neck muscles. However, the PREEMPT protocol uses superficial structures (e.g., the hairline) as landmarks, which results in inconsistencies. The postulated analgesic mechanism implies that the injection should be administered to sensory nerves rather than to muscles. 

The optimal intervention could be elucidated by focused observations of the nerve distribution in the target area based on consistent anatomical landmarks. The present study was therefore designed to determine the topographical site of the auriculotemporal nerve (ATN) as it exits from the deep layer to the superficial layer, and to propose the effective injection points for treating CM.

## 2. Results

The ATN appeared superficially in the anterosuperior region of the tragus. In that region, the nerve is positioned deeper than the superficial temporal artery and passes between the artery and the superficial temporal vein (Figure 1).

In the superficial layer, it is divided into anterior and posterior divisions. The anterior division runs in a superior direction, while the posterior division runs in front of the ear and the several branches are distributed to the skin. The area in which the distribution of the nerve was greatest is presented in Figure 2A.

## 3. Discussion

Nerve block procedures have been used in headache conditions as well as in neuropathic pain. Despite the limited evidence base, previous studies reported that the peripheral nerve block procedures including occipital, supraorbital, supratrochlear, and ATN can be effective in primary headaches [18]. ATN is the main source of sensory innervation in the temporal area. This nerve derives from the mandibular nerve and first appears in the deep layer, then becomes superficial in the temporal area near the temporomandibular joint. Determining the most appropriate area to inject BTX-A first requires the definition of consistent anatomical landmarks. The PREEMPT protocol uses the anterior border of the temporalis muscle and the hairline as landmarks, but this results in inconsistencies, since the hairline varies markedly between individuals. The present study therefore used a consistent reference line defined using the canthus and tragus, which also makes it easy to identify by visual inspection. The ATN appeared superficially in the anterosuperior region of the tragus. It then traveled superiorly and superficially in the temporal area within the subcutaneous tissue.

Baumel (1971) reported on the anatomy of the ATN [19]. This nerve is formed by two roots that arise from the posterior aspect of the undivided stem of the mandibular nerve. Six principal named branches are generally reported: two rami communicating with the facial nerve, two nerves to the external acoustic meatus, the anterior auricular nerve, and the superficial temporal ramus. The nerve targeted in the present study for BTX-A injection was the superficial temporal ramus. This is the largest, single terminal branch, which first travels laterally, usually supplying several branches to the temporomandibular articulation while passing posterior to it. Once becoming superficial, it turns superiorly and crosses the root of the zygomatic arch, where it lies posteromedial to the superficial temporal artery. 

Previous studies have found that injecting BTX-A results in it diffusing over an area with an average diameter of 6–10.9 mm [20,21,22]. We therefore suggest that the optimal BTX-A injection points for CM are in the temporal region (Figure 2B). The first point is about 2 cm anterior and 3 cm superior to the two orthogonal reference lines, the second point is about 4 cm superior to the first point, and the third and fourth points are both about 2 cm superior to the first point, but respectively 1 cm anterior and posterior to it. Because the ATN travels superficially over the temporalis muscle, the agent needs to be injected subcutaneously (Figure 3).

BTX-A is the only prophylactic therapy that has been approved specifically for CM. The first clinical observations of the possible beneficial effects of BTX-A on headache were made when the toxin was applied to patients for cosmetic reasons. This has resulted in the nomenclature of the injection sites being based on muscles. BTX-A injections for CM have evolved into a standardized technique covering at least 31 distinct injection sites across the head and neck regions [17]. An example guide for injecting in the temporal area is as follows: The patient is instructed to clench his or her teeth to assist in locating the anterior aspect of the temporalis muscle, which is then palpated. The first injection is made just behind this point, approximately two finger widths posterior to the hairline. The second injection is made approximately 0.5 cm superior and 1.5 cm posterior to the first injection in the medial aspect of the muscle. The third injection site is parallel and approximately 1.5 cm posterior to the second injection site. The fourth fixed-site injection is 1.5 cm below and perpendicular to the second injection, into the medial aspect of the muscle. 

In contrast, Ashkenazi and Blumenfeld (2013) suggested injection points for CM patients based on the consideration of the distribution of sensory nerves [23]. These can best be located by palpating the tragus of the ear, with the first injection being made about 3 cm vertically above the tragus, and a second injection made at least 1.5 cm higher. A third injection is made between these first two injection points, but about 1.5 cm anterior to them. The final injection into the temporalis muscle is made in line with the second injection, approximately 1.5 cm in a posterior direction.

In this study, we proposed the injection points into temporal region, based on the topographic anatomy of the ATN. However, we could not clarify whether the temporal fascia plays in role as a barrier of the BTX-A diffusion. A further study is needed about the diffusion and migration of BTX-A.

How BTX-A affects CM is not fully understood. However, the roles of BTX-A in atypical pain processing, central sensitization, cortical hyperexcitability, and neurogenic inflammation have been studied. BTX-A directly inhibits peripheral sensitization by attenuating the exocytosis of neuropeptides and neurotransmitters from peripheral sensory neurons, thereby indirectly reducing central sensitization [24]. The release of copious vasoactive neuropeptides such as CGRP and the resultant neurogenic inflammation might contribute to the pathophysiology of CM. Animal studies have provided evidence that BTX-A can block the stimulated release of CGRP, glutamate, and substance P from trigeminal neurons and inhibit the activation of second-order neurons within the spinal cord that are responsible for the transmission of pain signals [25,26,27]. During a migraine attack, peripheral sensitization occurs due to the activation of trigeminal nerve fibers innervating the dura mater and blood vessels. Neurogenic vasodilation of the dural blood vessels causes dural inflammation. The BTX-A -induced suppression of experimental dural neurogenic inflammation was found to be mediated by the axonal transport of the toxin within the trigeminal nerve in an animal model [28]. In addition, the peripheral delivery of BTX-A may result in it being transported axonally to the central spinal trigeminal nucleus caudalis (TNC) [29]. The presence of cleaved SNAP-25 in the TNC suggests that BTX-A also acts at the level of second-order neurons in the TNC, which receive convergent nociceptive input from the trigeminal nerve and mediate central sensitization [28,29]. Matak and Lackovic (2014) and Pellett (2015) have reviewed the analgesic effects of BTX-A [30,31]. BTX-A might affect nociceptive processing in several ways, at the levels of peripheral nerves, dorsal root ganglion, spinal dorsal horn, and even the brainstem. 

The analgesic effects exerted by BTX-A via effects on sensory neurons as discussed above indicate that injections of this agent should target sensory nerves in the trigeminal–occipital–cervical complex. The ATN is the main source of innervation in the temporal area. 

## 4. Conclusions

BTX-A should be injected superficially into the temporal area in the ATN distribution area, under guidance based on clearly identifiable and consistent anatomical landmarks and a reference axis. We suggest that the optimal BTX-A injection points for CM are in the temporal region. The first point is about 2 cm anterior and 3 cm superior to the two orthogonal reference lines based on the canthus and tragus, the second point is about 4 cm superior to the first point, and the third and fourth points are both about 2 cm superior to the first point, but respectively 1 cm anterior and posterior to it. Because the ATN travels superficially over the temporalis muscle, the agent needs to be injected subcutaneously.

## 5. Materials and Methods

This study investigated ATNs on 36 hemifaces of 25 Korean cadavers (17 males and 8 females with a mean age of 76.6 years); 14 specimens were excluded due to loss during dissection. The 25 cadavers were embalmed with a fluid containing 10% phenol/formaldehyde. After removing the skin and subcutaneous tissues of the upper and lower faces, the anatomical structures of the ATN were investigated, focusing on the temporal region. The superficial temporal fascia was carefully reflected so that the ATN distribution layers could be discerned. Each specimen was dissected in detail while taking extreme care not to damage the main structures. 

After dissection, the canthus and tragus were set as landmarks to establish a horizontal reference line. A vertical reference line was then set at the front of the tragus, perpendicularly to this horizontal line. A right-angle ruler was positioned along these reference lines, and photographs were taken. The obtained photographs were calibrated to the same scale using computer graphics software (Photoshop, Adobe, USA). All photographs were adjusted to the right side, and ATNs were traced on each specimen. The photographs were then superimposed on a layer-by-layer basis, and the area with the greatest nerve distribution was marked.

## Figures and Tables

**Figure 1 toxins-12-00214-f001:**
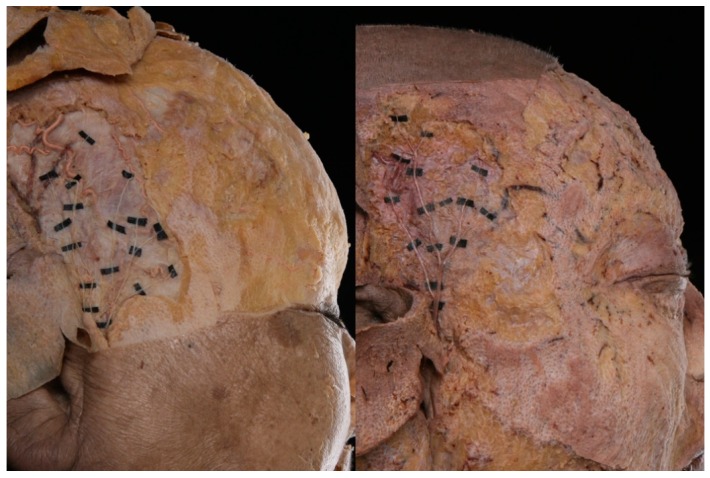
Photographs showing the superficial layer of the auriculotemporal nerve (ATN) distribution in two representative specimens.

**Figure 2 toxins-12-00214-f002:**
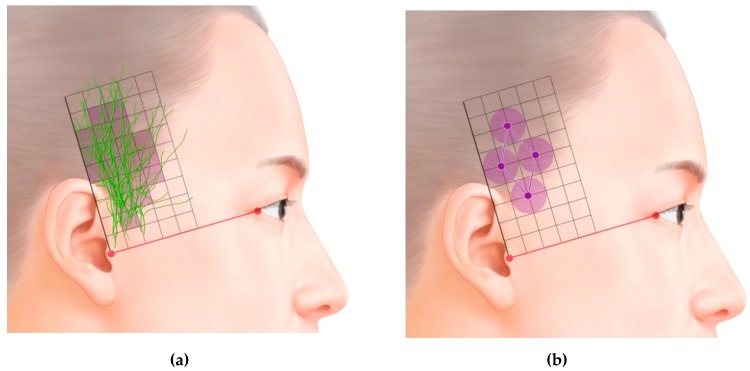
(**A**). Superimposed traces following ATNs. (**B**). Optimal sites for injecting botulinum toxin type-A (BTX-A) into the temporal region based on the area of the ATN, as revealed in this study (grid = 1 cm × 1 cm).

**Figure 3 toxins-12-00214-f003:**
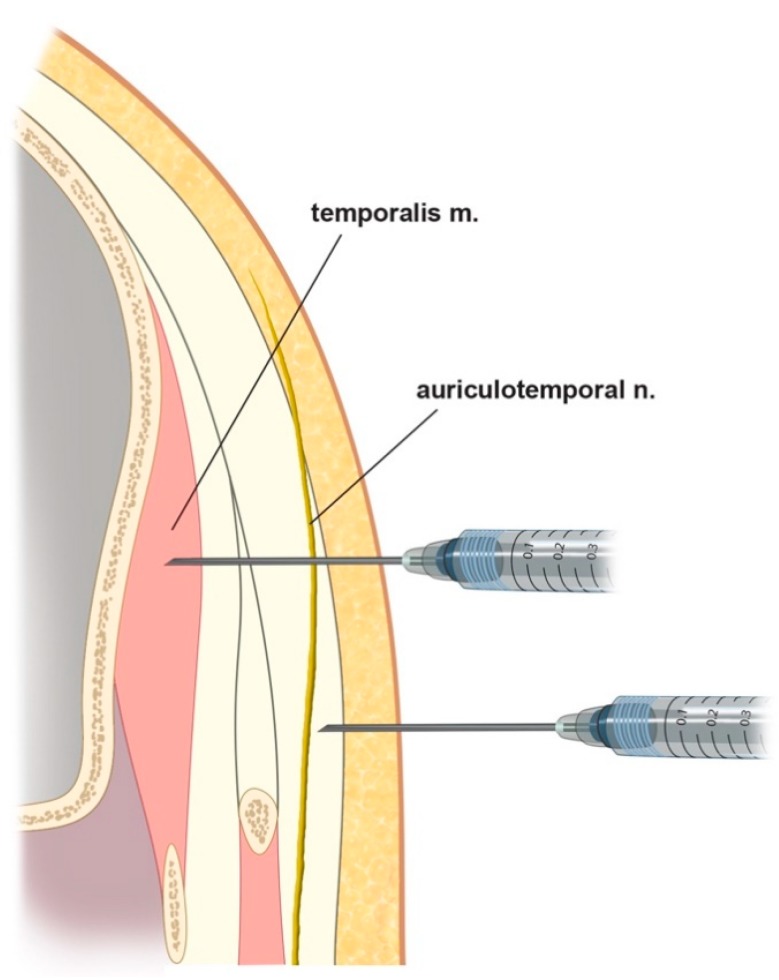
Schematic comparing different injection methods: upper, intramuscular injections for muscle targets (e.g., sleep bruxism); lower, subcutaneous injections for sensory nerve targets (e.g., chronic migraine (CM) and nerve block), as suggested from this study.

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
