# Peer review of "A Proposal for Botulinum Toxin Type A Injection Into the Temporal Region in Chronic Migraine Headache"

_toxins, 2020, doi:10.3390/toxins12040214_

Round 1
Reviewer 1 Report
Interesting study with novel anatomical material relevant to injection protocols for migraine. I have some comments and questions for the authors.
- Line 2: I think you mean to inject in the vicinity of the sensory nerves, not the nerves themselves?
- Lines 36-40: There is very good evidence that the analgesic effects of BTXA are related to inhibition of muscle spasm or activity in some conditions, including cerebral palsy which the authors do not list. Given the ability of BTXA to diffuse across fascial barriers, the authors proposed protocol and injection technique will unfortunately not solve this key question. Injection below the fascia will inhibit contraction of the temporalis muscle but some toxin will diffuse outward to the branches of the ATN. Conversely, injection superficial to the fasica will still allow diffusion of BTXA below the fascia and result in chemodennervation of the temporalis muscle. I think the duality of action of BTXA in this area must be stressed. This should also be listed by the authors as a study limitation to their hypothesis.
- Lines 46 and 47: their may be dual actions: muscle and sensory effects combined.
- The figures are of excellent standard but the fact that the temporalis fascia is not an effective block to the diffusion of BTXA must be indicated.
- Line 150: is incomplete "be injected...."
- Given that the central hypothesis for the study is the sensory effects of BTXA and the suggested revision of injection protocols, the reference by Levin 2010 Neurotherapeutics on the use of local anaesthetic blocks in headache should be included please.
Author Response
- Line 2: I think you mean to inject in the vicinity of the sensory nerves, not the nerves themselves?
Response: Yes. It means that the injection points are sensory nerve distribution area not direct into nerve.
- Lines 36-40: There is very good evidence that the analgesic effects of BTXA are related to inhibition of muscle spasm or activity in some conditions, including cerebral palsy which the authors do not list. Given the ability of BTXA to diffuse across fascial barriers, the authors proposed protocol and injection technique will unfortunately not solve this key question. Injection below the fascia will inhibit contraction of the temporalis muscle but some toxin will diffuse outward to the branches of the ATN. Conversely, injection superficial to the fasica will still allow diffusion of BTXA below the fascia and result in chemodenervation of the temporalis muscle. I think the duality of action of BTXA in this area must be stressed. This should also be listed by the authors as a study limitation to their hypothesis.
Response: We agree that there should be denervation of temporalis muscle; however, its contribution to preventive efficacy of chronic migraine is currently not recognized for migraine prevention. However, we added this potential effect, i.e. the role of temporalis fascia in the relevant paragraph as a limitation of this study.
Lines 122 to 124: In this study, we suggested proposal injection points which is based on topographic anatomy. However, we could not clarify whether the temporal fascia plays in role as a barrier of the BoNT diffusion. A further study is needed about the diffusion and migration of BoNT in this area.
- Lines 46 and 47: their may be dual actions: muscle and sensory effects combined.
Response: The efficacy of muscle denervation of pericranial muscles are not well accepted as the mechanism of migraine prevention because “muscular origin” is not considered as an important mechanism for migraine compared to that for tension-type headache. (American Academy of Neurology (AAN) practice guideline update, “Botulinum neurotoxin for the treatment of blepharospasm, cervical dystonia, adult spasticity, and headache,” which was published in Neurology®, 2016)
- The figures are of excellent standard but the fact that the temporalis fascia is not an effective block to the diffusion of BTXA must be indicated.
Response: Yes, we put the temporal fascia story in the discussion as limitation of this study.
- Line 150: is incomplete "be injected...."
Response: ……..be injected subcutaneously.
- Given that the central hypothesis for the study is the sensory effects of BTXA and the suggested revision of injection protocols, the reference by Levin 2010 Neurotherapeutics on the use of local anaesthetic blocks in headache should be included please.
Response: Yes. We put Dr. Levin’s nerve block article as a reference in the discussion.
Nerve block procedures have been used in headache conditions as well as in neuropathic pain. Despite the limited evidence base, previous studies reported the peripheral nerve block procedures including occipital, supraorbital, supratrochlear and ATN can be effective in primary headaches.

Reviewer 2 Report
Dear authors;
The subject of the article is very interesting and related to the common neurological problem of chronic migraine. The article was written in a correct and comprehensive language, the English is understandable, and the results provide an advance in current knowledge. The data and analyses are presented appropriately. In my opinion the highest standards for presentation of the results are used and the conclusions are interesting for the readership of the Journal. Additionally the work provide an advance towards the current knowledge. The results of the conducted tests were presented correctly and clearly. The literature review is rich and closely related to the subject of the article. The figures in the article help to understand the complexity of research.
I suggest small changes in the order of the paragraphs; Introductions, Material and methods, Results, Discussion and Conclusion.
Author Response
The order of each section in this Toxins is different from other journals; material & method is located in rear part.